# CFD Modeling of UV/H$_2$O$_2$ Process in Internal Airlift Circulating Photoreactor

**Minghan Luo [1,2] , Fan Zeng [1], Taeseop Jeong [3,\*], Gongde Wu [1,2] and Qingqing Guan [1]**

[1] School of Environmental Engineering, Nanjing Institute of Technology, Nanjing 211167, China; leon96201@163.com (M.L.); zf85@njit.edu.cn (F.Z.); wugongde@njit.edu.cn (G.W.); guanqingqing2@163.com (Q.G.)

[2] Energy Research Institute, Nanjing Institute of Technology, Nanjing 211167, China

[3] Department of Environmental Engineering, Chonbuk National University, Chonbuk 561-756, Korea

\* Correspondence: jeongts@jbnu.ac.kr

**Abstract:** UV chemical degradation is a low-cost and sustainable wastewater treatment technology that protects the environment. In this study, computational fluid dynamics (CFD), mass transfer, and photochemical kinetic models combined with the continuous flow mode of UV/H$_2$O$_2$ were applied for the photochemical reaction of internal airlift circulation photocatalytic reactor to improve the efficiency of the reaction. Results show that with the increase in gas flow rate, the turbulence intensity and internal circulation effect of liquid in the reactor can be enhanced under the condition of constant baffle spacing. The CFD simulation prediction results of the chemical components in the liquid flow show that H$_2$O$_2$ has a high correlation with the OH radical formation, which depends on the intensity of UV. Thus, the degradation rate of methylene blue (MB) has a high correlation with UV intensity. The degradation efficiency of MB is improved with the increase in gas velocity by comparing the experimental data with the numerical simulation data. The experimental data are generally lower than the numerical prediction data, and although a certain difference range is observed, the overall prediction results are better.

**Keywords:** UV/H$_2$O$_2$; photoreactor; CFD; water treatment; MB

## 1. Introduction

Advanced oxidation processes (AOPs) is a cost-effective solution for removal of harmful and refractory pollutants from water. This method is preferred and ideal in the field of water treatment and air purification. The hydroxyl radical produced in AOPs, which is the most aggressive substance without choice, has a strong oxidation potential of 2.80 v. UV-based AOP technology has been widely used in the field of drinking water disinfection and has made important progress and achieved commercialization. However, in the field of wastewater treatment, cases of applying this method are few, because the reaction system needs a long reaction time and satisfies the basic conditions, such as sufficient reaction among the components in the reactor. In addition, the structural design of this reaction system also needs the design technology of chemical reaction mechanism and kinetic rate constant for optimization. At present, most large-scale photocatalytic reactors use the air lift method to enhance the mixing effect among the components and reduce the dead zone in the reactor [1,2]. The air lift internal circulation photoreactor uses air as the driving force to realize the internal circulation flow of water in the reactor. This photoreactor can realize the full mixing of substances in the fluid without mechanical stirring and pump lifting. It is widely used in various fields, such as in biology, environment, and chemical industry, because of its simple structure and low energy consumption [3,4].

Photochemical AOPs use UV radiation in combination with a chemical species (such as hydrogen peroxide, ozone, persulfate) to generate the hydroxyl radicals [5]. In many AOPs technologies, UV/$H_2O_2$ process is one of the more promising technologies, that in $H_2O_2$ absorbs light energy and breaks the O–O bond to form a strong oxidant OH. The reaction rate constant of $\cdot$ OH with most organic compounds can reach $10^6$–$10^{10}$ $M^{-1}s^{-1}$, which can oxidize and decompose toxic and refractory organic pollutants effectively. The UV/$H_2O_2$ system is most often used to degrade organic compounds by the combined mechanisms of direct UV photolysis and hydroxyl radical reactions [5]. Therefore, UV/$H_2O_2$ process has been widely investigated and applied because of its strong oxidation capacity and clean environmental protection advantages. It has gradually become the preferred method for treating refractory organics and is a promising water treatment technology [6,7]. However, different target pollutants, water quality, concentration of $H_2O_2$, and intensity of ultraviolet light have great differences on the degradation performance of organic compounds. In addition, the geometry of the reactor, the hydrodynamic characteristics in the reactor, and the working conditions of the UV system determine the distribution of UV light.

Numerical simulation based on computational fluid dynamics (CFD) has been widely used in various reactor flow field research because of the development of computer technology, numerical method, and hydrodynamics theory. However, due to the complexity of the multiphase flow process, the research mainly focuses on the Reynolds averaged hydrodynamics of the reactor, which is very limited for understanding the local hydrodynamic conditions. The application of UV/AOPs technology in airlift reactor can increase the contact area of gas and liquid reactants and increase the effective light area in the reactor. Therefore, it can greatly improve the kinetic process of photocatalytic reaction, and has the advantages of strong mass transfer ability and controllable hydraulic retention time. Minghan Luo et al. used the multiphase flow method to simulate the hydrodynamics of an internal airlift circulation photocatalytic reactor, and obtained the information of phase holdup, fluid flow characteristics, and mixing effect [8]. However, the application of UV/AOPs technology combined with chemical reaction kinetics model, radiation model, and pollutant transport model has not been reported. In addition, only few authors have explored the degradation characteristics of the airlift internal circulation photocatalytic reactor through experiments [9,10]. However, when the catalyst is added to the catalytic oxidation reaction process of the internal circulation photocatalytic reactor, the hydrodynamic and chemical reaction situations are still unclear.

Although some numerical methods have been developed to study the performance of UV/AOPs technology for conventional reactors, they are limited by the need to combine the hydrodynamic model and the chemical kinetic model. Design of UV/$H_2O_2$ process requires knowledge about both system configuration and chemical kinetics [11]. Therefore, when UV AOPs technology is applied to different reactors, more appropriate numerical tools are needed for the preliminary design and optimization. B.A. Wols et al. used the CFD model combined with ultraviolet radiation, transmission, and photochemical models to calculate the degradation of pollutants in the UV system, and evaluated the transport of pollutants by Lagrange method and Euler method, respectively. The research results show that these two methods can be used for the degradation process of UV/$H_2O_2$ system [5]. Scott M. et al. [11] used the CFD model of the UV/$H_2O_2$ system to compare CFD model characteristics (e.g., ultraviolet radiation submodel, turbulent submodel, and kinetic rate constant) with system parameters (e.g., inlet flow rate, hydrogen peroxide capacity, and radical scavenger sensitivity analysis). They found that the CFD model predicted the experimental percent removal of methylene blue. Moreover, its degradation was sensitive to the background dissolved organic carbon concentration. The reaction mechanisms for the degradation of organic contaminants by UV/AOPs processes typically comprises a complex chain of fast chemical reactions [11]. Therefore, intermediates and products that resulted from these processes can be very sensitive to UV irradiation rates in the reactor, as well as turbulence and mixing levels [8]. Wojciech et al. [6] studied the radiation doses received in different zones of a reactor according to the spatial variability of the velocity field and radiation intensity.

Compared with the actual experiment, the numerical simulation has lower cost and shorter cycle. Thus, the fluid flow process is presented vividly, and data that are difficult to be observed in actual experiments are obtained. In recent years, researchers have performed the numerical simulation of fluid dynamics and the chemical reaction of various reaction devices [6,12,13], whereas others have carried out corresponding experimental verification on the simulation results [11], which proves that the numerical calculation method is very feasible in simulating the hydrodynamic characteristics in the reactor. In this study, UV/H$_2$O$_2$ photochemical reaction system was applied to the airlift internal circulation reactor. First, the velocity and volume fraction distribution of gas and liquid and the turbulence intensity were investigated. Second, the chemical reaction kinetic model was used to simulate the migration and transformation of chemical components in the air lift internal circulation photoreactor. Finally, the experimental data were compared to provide theoretical support for the design and application of this kind of reactor in the future.

## 2. Mathematical Methods and Materials

### 2.1. Hydrodynamic, Mass Transfer, and Turbulent Model

Based on the principles of conservation of mass and momentum, a two-dimensional CFD model was developed to calculate the local hydrodynamics in photoreactor, and the phasic volume fraction $\alpha$ must satisfy the relation:

$$\sum_{i=1}^{n} \alpha_i = 1 \tag{1}$$

where $n$ is the total number of phases; the subscript $i$ represents gas or liquid phase. The conservation equations are written by performing an ensemble average of the local instantaneous balance for each phase. The motion of each phase is governed by corresponding mass and momentum conservation equations.

Continuity equations:

$$\frac{\partial(\alpha_i \cdot \rho_i)}{\partial t} + \nabla \cdot (\alpha_i \rho_i \vec{u}_i) = 0 \tag{2}$$

where $\alpha$, $\rho$, and $\vec{u}$ stand for the volume fraction, density, and velocity vector, respectively.

Momentum equation:

$$\frac{\partial(\alpha_i \rho_i \vec{u}_i)}{\partial t} + \nabla \cdot (\alpha_i \rho_i \vec{u}_i \vec{u}_i) = -\alpha_i \nabla P_i + \nabla \cdot (\alpha_i \mu_i (\nabla \vec{u}_i - (\nabla \vec{u}_i)^T)) + \alpha_i \rho_i \vec{g} \pm \vec{F}_i \tag{3}$$

where $P$, $\mu$, and $\vec{g}$ are the pressure, viscosity, and gravity acceleration, respectively. $\vec{F}_i$ is the interfacial force acting on phase $i$ due to the presence of the other phase, $j$.

The turbulent dispersion force is the result of the turbulent fluctuations of liquid velocity. The standard $k - \varepsilon$ model for single-phase flows has been extended for the two-phase flows, which can be described as:

$$\frac{\partial}{\partial t}(\alpha_l \rho_l k_l) + \frac{\partial}{\partial x_i}(\alpha_l \rho_l \vec{u}_l k_l) = \frac{\partial}{\partial x_i}\left[\alpha_l\left(\mu_l + \frac{\mu_{tl}}{\sigma_k}\right)\frac{\partial}{\partial x_i}k_l\right] + \alpha_l \rho_l - \alpha_l \rho_l \varepsilon_l \tag{4}$$

$$\frac{\partial}{\partial t}(\alpha_l \rho_l \varepsilon_l) + \frac{\partial}{\partial x_i}(\alpha_l \rho_l \vec{u}_l \varepsilon_l) = \frac{\partial}{\partial x_i}\left[\alpha_l\left(\mu_l + \frac{\mu_{tl}}{\sigma_\varepsilon}\right)\frac{\partial}{\partial x_i}\varepsilon_l\right] + \alpha_l \frac{\varepsilon_l}{k_l}(C_{\varepsilon 1 p_l} - C_{\varepsilon 2 \rho_l \varepsilon_l}) \tag{5}$$

where $C_{\varepsilon 1}$, $C_{\varepsilon 2}$, $\sigma_k$, and $\sigma_\varepsilon$ are parameters in the standard $k - \varepsilon$ model and the following values are selected: $C_{\varepsilon 1} = 1.44$, $C_{\varepsilon 2} = 1.92$, $\sigma_k = 1.0$, and $\sigma_\varepsilon = 1.3$. In addition, the turbulent viscosities $\mu_{tl}$ can be computed by other equations [8,14].

## 2.2. Interphase Force Models

During momentum exchange between the different phases, only the drag and turbulent dispersion force was considered [8,15]. Accordingly, the interfacial force $\vec{F}_i$ is approximated to be the drag force and interphase turbulent dispersion force:

$$\vec{F}_i = \vec{F}_i^D + \vec{F}_i^T \tag{6}$$

where $\vec{F}_i^D$ $and$ $\vec{F}_i^T$ are the drag force and turbulent dispersion force, respectively.

The drag force between the gas and liquid phases is calculated by the following equation:

$$\vec{F}_{lg}^D = C_{D,lg} \frac{3}{4} \frac{\rho_l \alpha_g}{d_g} \left| \vec{v}_g - \vec{v}_l \right| \left( \vec{v}_g - \vec{v}_l \right) \tag{7}$$

where $C_D$ is the drag coefficient. The Grace relation is chosen for the drag force coefficient because bubbles are experimentally observed to be spherical and dispersed [15].

The turbulent dispersion force, $\vec{F}_i^T$, is calculated by the model of Lopez de Bertodano [16]:

$$\vec{F}_i^T = C_{TD} C_D \frac{v_{tl}}{\sigma_{tl}} \left( \frac{\nabla \alpha_g}{\alpha_g} - \frac{\nabla \alpha_l}{\alpha_l} \right) \tag{8}$$

where $C_{TD}$ is the momentum transfer coefficient for the interphase drag force, $C_D$ is the drag coefficient as described above, $v_{tl}$ stands for turbulent viscosity, and $\sigma_{tl}$ is the liquid turbulent Schmidt number. $\alpha_g$ and $\alpha_l$ are the gas and liquid phase volume fractions, respectively.

## 2.3. Radiative Transfer Model

The radiative transfer equation (RTE) for an absorbing, emitting, and scattering medium at position $\vec{r}$ in the direction $\vec{s}$ is

$$\frac{dI(\vec{r}, \vec{s})}{ds} + (a + \sigma_s) \, I(\vec{r}, \vec{s}) = an^2 \frac{\sigma T^4}{\pi} + \frac{\sigma_s}{4\pi} \int_0^{4\pi} I(\vec{r}, \vec{s}') \, \varnothing(\vec{s}, \vec{s}') \, d\Omega' \tag{9}$$

where $\vec{r}$ and $\vec{s}$ are position and direction vectors, respectively. The $I$ is the radiation intensity, which depends on position and direction, $n$ is the refractive index, $\sigma$ is the Stefan–Boltzmann constant ($5.67 \times 10^{-8}$ Wm$^{-2}$K$^{-4}$), $\alpha$ is the absorption coefficient, $\sigma_s$ is the scattering coefficient, $\varnothing$ is the phase function, and $\Omega\prime$ is the solid angle. Additionally, $(a + \sigma_s)s$ is the optical thickness or opacity of the fluid. The refractive index $n$ is important when considering radiation in semitransparent media [17].

## 2.4. Kinetic Model

In this study, photolysis of methylene blue (MB) was selected as a model reaction material. Since methylene blue is not directly photolyzed and does not react with $H_2O_2$ alone, MB is a dye that is good indicator for evaluation in the UV/AOPs study due to the decolorization upon reaction with the ·OH. The detailed kinetic model for the UV/$H_2O_2$ systems has been extensively verified by several previous studies and particularly in [18] for continuous flow stirred tank reactors and by Scott M. Alpert et al. [11].

Table 1 summarizes the system of kinetic rate equations describing the production of the ·OH from $H_2O_2$ and the corresponding degradation of MB, along with their rate constants. The photolysis of $H_2O_2$ is shown as reaction in Table 1. In this study, the kinetic model was incorporated into CFD model using User-Defined Function (UDF) of FLUENT to prescribe the net rate of generation via $R_i$ for each chemical species considered (Equations (1)–(7)).

**Table 1.** Kinetic model of UV/$H_2O_2$ advanced oxidation of MB.

| No. | Reaction Equation | Rate Constant, $M^{-1}s^{-1}$ | Reference |
|---|---|---|---|
| 1 | $H_2O_2 + hv \rightarrow 2 \cdot OH$ | $\varnothing_{H_2O_2} = 0.5$ $\varepsilon_{H_2O_2} = 19.6\ M^{-1}cm^{-1}$ | Alpert et al. [11] |
| 2 | $H_2O_2 + \cdot OH \rightarrow H_2O + HO_2 \cdot (\rightarrow H^+ + \cdot O_2^-)$ $R_{H_2O_2} = -k_1[H_2O_2][\cdot OH]$ $R_{\cdot OH} = -k_1[H_2O_2][\cdot OH]$ $R_{\cdot O_2^-} = k_1[H_2O_2][\cdot OH]$ | $k_1 = 2.7 \times 10^7$ | Alpert et al. [11] |
| 3 | $H_2O_2 + \cdot O_2^- \rightarrow \cdot OH + O_2 + OH^-$ $R_{H_2O_2} = -k_2[H_2O_2]\left[\cdot O_2^-\right]$ $R_{\cdot O_2^-} = -k_2[H_2O_2]\left[\cdot O_2^-\right]$ $R_{\cdot OH} = k_2[H_2O_2]\left[\cdot O_2^-\right]$ | $k_2 = 0.13$ | Alpert et al. [11] Alpert et al. |
| 4 | $\cdot OH + \cdot OH \rightarrow H_2O_2$ $R_{\cdot OH} = -k_3[\cdot OH][\cdot OH]$ $R_{H_2O_2} = k_3[\cdot OH][\cdot OH]$ | $k_3 = 5.5 \times 10^9$ | Alpert et al. [11] |
| 5 | $OH + \cdot O_2^- \rightarrow O_2 + OH^-$ $R_{\cdot OH} = -k_4[\cdot OH]\left[\cdot O_2^-\right]$ $R_{\cdot O_2^-} = -k_4[\cdot OH]\left[\cdot O_2^-\right]$ | $k_4 = 7.0 \times 10^9$ | Alpert et al. [11] |
| 6 | $OH + MB \rightarrow Products$ $R_{\cdot OH} = -k_{MB,OH}[\cdot OH][MB]$ $R_{MB} = -k_{MB,OH}[\cdot OH][MB]$ | $k_{MB,OH} = 2.1 \times 10^{10}$ | Alpert et al. [11] |
| 7 | $HCO_3^- + \cdot OH \rightarrow CO_3^- + H_2O$ $R_{\cdot OH} = -k_{HCO_3^-,OH}[\cdot OH]\left[HCO_3^-\right]$ | $k_{HCO_3^-,\cdot OH} = 8.5 \times 10^6$ | Alpert et al. [11] |
| Composite | $R_{\cdot OH} = 2\varnothing_{H_2O_2}\varepsilon_{H_2O_2}[H_2O_2]E'_{CFD} -$ $k_1[H_2O_2][\cdot OH] + k_2[H_2O_2]\left[\cdot O_2^-\right] -$ $k_3[\cdot OH][\cdot OH] - k_4[\cdot OH]\left[\cdot O_2^-\right] -$ $k_{MB,OH}[\cdot OH][MB] - k_{HCO_3^-,OH}[\cdot OH]\left[HCO_3^-\right]$ $R_{\cdot O_2^-} = k_1[H_2O_2][\cdot OH] - k_2[H_2O_2]\left[\cdot O_2^-\right] -$ $k_4[\cdot OH]\left[\cdot O_2^-\right]$ $R_{H_2O_2} = -\varnothing_{H_2O_2}E'_{CFD}\varepsilon_{H_2O_2}[H_2O_2] -$ $k_1[H_2O_2][\cdot OH] - k_2[H_2O_2]\left[\cdot O_2^-\right] +$ $k_3[\cdot OH][\cdot OH]$ $R_{MB} = -k_{MB,\ OH}[\cdot OH][MB]$ | | |

## 2.5. Airlift Photoreactor Setup, Geometry, and Grid Generation

A flow through that continuously operated the airlift photoreactor was used to evaluate the CFD results experimentally. A simple two-dimensional geometry used for the internal airlift circulating photoreactor is shown in Figure 1. It is conical bottomed, internally irradiated with the light source being placed in the central core axis, and has a draft tube with the internal circulation occurring around the light source. The reactor is divided into the central circulation area, the bottom air, and water distribution area. It is operated with a 54 W low-pressure mercury UV lamp (customized) that was approximately 2.3 cm in diameter and 50 cm in length and longitudinally placed at the axial center of the reactor. In this study, the position of the draft tube was set to 0.4 of r/R, where r is the distance between the reactor wall and the draft tube, and R is the distance between the reactor wall and the light source (R) [8]. The inner part of the reactor is composed of a 60 cm long draft tube and a 50 cm long UV lamp. During the operation of the reactor, the solution with MB and $H_2O_2$ was injected from the bottom of the reactor at a rate of 0.3 m/s continuously. At the same time, the air was injected at flow velocity of 0.01, 0.04, 0.07, and 0.10 m/s, respectively. The reactor reached a stable condition after 120 min, then a total of six samples were taken at 30 min time intervals for MB determination. In addition, in order to further analyze the data of various physical elements in the reactor, the data

used in this study are Section 1 in the middle of the reactor and Section 2 at the outlet of the reactor shown in Figure 1.

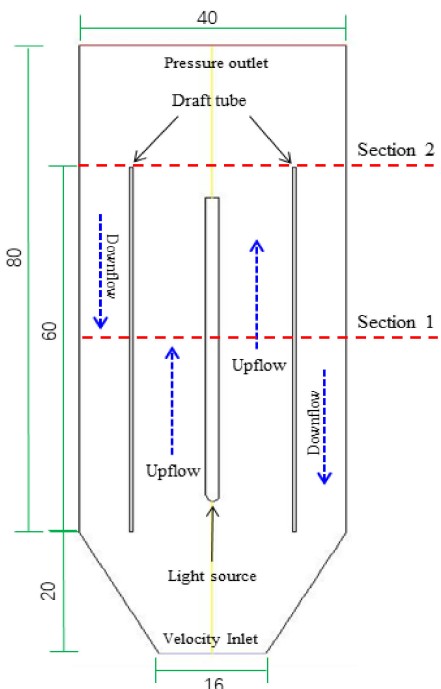

**Figure 1.** Schematic diagram of the internal airlift circulating photoreactor for the simulation (cm).

The geometry was created inDesignModeler software (ANSYS, Inc., Canonsburg, PA USA). Given the characteristics of the axially symmetric structure of the reactor, the flow field simulation can be simplified reasonably by a 2D axial symmetry model. An unstructured numerical grid was implemented with a total number of 14,462 elements, where the minimum size of the grid is 0.15 mm, and the maximum face size is 5.0 mm.

### 2.6. Initial and Boundary Conditions

For the initial conditions, the liquid and gas phases enter the reactor from the bottom of the reactor at different flow rates. The gas and liquid phases were air and water in the standard state, respectively. A uniform air velocity $U_g$ (0.01, 0.04, 0.07, and 0.10 m/s) was simulated from the gas inlet. Meanwhile, the liquid velocity was set at 0.3 m/s, the inlet MB concentration as model contaminant equaled 0.5 ppm, and the inlet $H_2O_2$ concentration was set equal to 10 ppm. The diameter and viscosity of air considered are 0.2 mm and $1.003 \times 10^{-3}$ Pa, respectively. The refractive index and absorption coefficient were assigned at 1.376 and 12.78 ($m^{-1}$, UVT = 88%), respectively.

A pressure boundary condition was applied to the top of the reactor with an average static reference pressure of 0 Pa. In addition, the periphery and inner area were defined as the opening and degassing boundary conditions, where the gas and liquid phases could leave the computation domain. For the reactor walls and the light source boundary, the gas phase was treated as free-slip, whereas a no-slip boundary condition was applied in the liquid phase [19]. A no-slip boundary condition was imposed on the walls. In addition, the zero diffusive flux of species was specified at the walls. As per the radiation field boundary conditions, the lamp radiation was defined as a zero-thickness semitransparent wall and no-reflecting walls.

### 2.7. Numerical Solution

ANSYS Fluent 16.2 was used to read the mesh and perform the CFD computations. The segregated steady state solver was used to solve the governing equations. Second-order upwind discretization

scheme was applied, except for pressure, for which the standard scheme was selected. The SIMPLE algorithm was selected for the pressure–velocity coupling. The variation of velocity magnitude, model contaminant concentration, and irradiation flux at several points of the computational domain were used as indicators of convergence (at least 20 iterations). Additionally, the convergence of the numerical solution was assured by monitoring the scaled residuals to a criterion of at least $10^{-4}$ for the MB concentration. Although simulation is always tracked with time, the solution algorithm has been run with steady and transient flow simulations.

### 2.8. Chemicals and Analytical Methods

For the airlift photoreactor experiments, the chemicals used for experiments were reagent grade or higher and supplied by Sigma–Aldrich. MB powder and $H_2O_2$ solution (30% *w/v*) were used as purchased without further purification. The MB and $H_2O_2$ were diluted separately with ultrapure laboratory water. Distilled water was used in all experiments and analytical determinations. The MB concentration in the airlift photoreactor effluent was determined spectrophotometrically by following the peak at 664 nm using a UV spectrophotometric probe (UV1800, Shimadzu Co., Kyoto, Japan spectrophotometer). The relationship between absorbance and concentration of MB solution is $y = 0.189x + 0.003$, where $R^2 = 0.9998$. The hydrogen peroxide concentration was determined via UV spectrophotometry by using $I_3^-$ method [20].

## 3. Results and Discussion

### 3.1. Hydrodynamics

The internal airlift circulation photoreactor uses air as the driving force to enable the liquid to circulate continuously in the reactor without mechanical stirring and water pump to achieve the purpose of full mixing. Figure 2 shows the velocity distribution of liquid phase in the internal airlift circulation photoreactor.

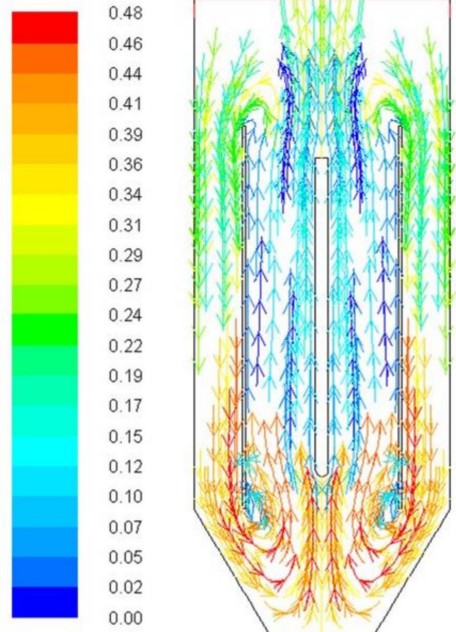

**Figure 2.** Liquid-phase velocity vector in vertical plane of the internal airlift circulation photoreactor (m/s).

The velocity distribution trend of the liquid phase in the entire photoreactor is evident. The gas enters the photoreactor from the entrance constantly, so that the local liquid density is lower than the surrounding liquid. Then, the gas rises and flows along the light source surface located in the middle

position until the outlet of the reactor is reached. Finally, the gas leaves the reactor. The surrounding high-density liquid flows to the inlet along the inner wall of the reactor from the outside of the draft tube to the inlet in time for the increase in liquid caused by the gas. Subsequently, this liquid rises and flows through the gas. This repeated flow enables the reactor to form a continuous internal circulation flow.

This kind of airlift internal circulation reactor can enable the liquid in the reactor to circulate repeatedly and achieve the purpose of fully mixing the components in the liquid effectively. This flow state is conducive to the effective chemical reaction among the components in the liquid. In other words, the fully mixed components in the liquid can be evenly irradiated by the light source in the photoreactor to make the reaction complete and increase the contact opportunities of various substances in the reactor.

### 3.2. Turbulence, Velocity, and Holdup

The radial data (Section 1 in Figure 1) at the key position of the middle height of the reactor were selected to describe the turbulent intensity (Figure 3) and the distribution of liquid velocity and gas holdup (Figure 4) in the reactor quantitatively under different gas velocities. In Figure 3, a and b correspond to the contours and radial distribution data of turbulence intensity at different gas velocities in the middle of the reactor, respectively. Combined with a and b diagrams, the turbulence intensity in the reactor increased with the gas velocity and is more obvious in the rising region of the reactor. The turbulence intensity in the bubble concentration region is higher in the rising region, which is close to the experimental results of Sokolichin et al. [21]. The reason is that the upward flow along the lamp source surface in the center of the reactor is fast when the velocity of the gas entering the reactor at the bottom of the reactor is high. The turbulence intensity is the highest when the gas velocity at the inlet is 0.10 m/s. However, 0.01 and 0.04 m/s have a huge difference. The large turbulence intensity can enable the material components in the liquid to be fully mixed and evenly irradiated by the light source effectively. These characteristics will be very conducive to the degradation efficiency of pollutants in the liquid.

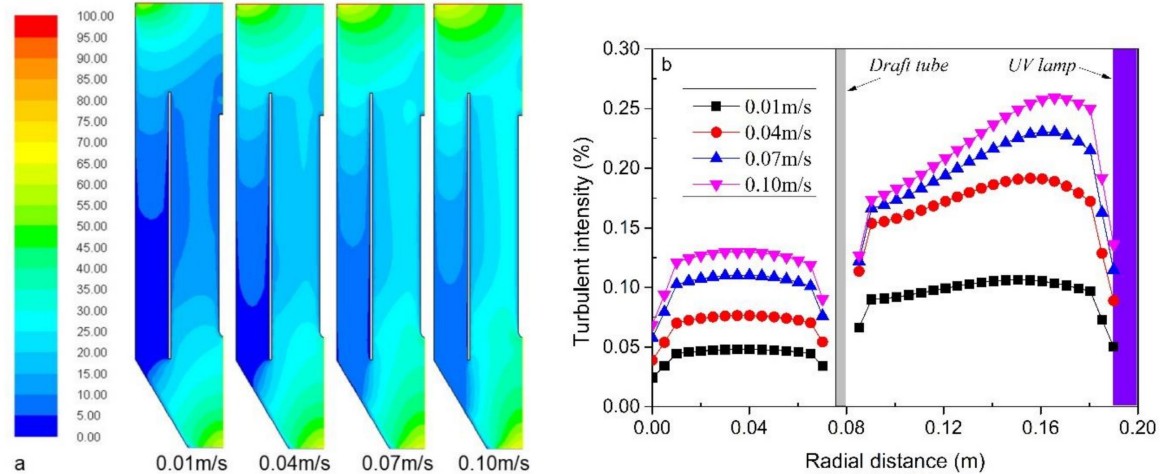

**Figure 3.** The contours (**a**) and the radial distribution (**b**) of turbulent intensity at different gas inlet velocities in Section 1.

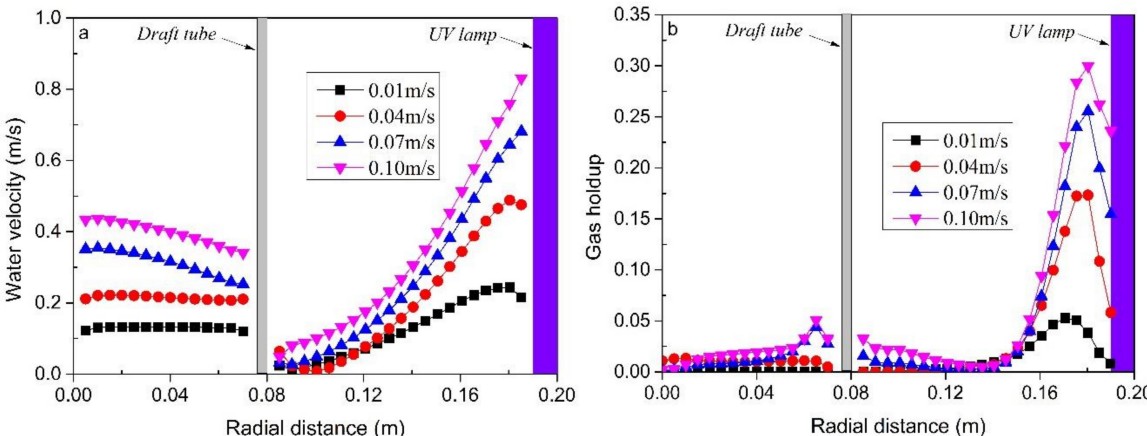

**Figure 4.** The radial distribution of water velocity (**a**) and gas holdup (**b**) at different gas inlet velocities in Section 1.

In Figure 4, a and b correspond to the radial distribution data of liquid velocity and gas holdup at different gas velocities in the middle of the reactor, respectively. Combined with diagrams a and b, the liquid velocity distribution and gas holdup tend to be higher near the light source and gradually decrease to the reactor wall with the increase in gas velocity at the bottom inlet of the reactor. In addition, the liquid velocity in the downflow region also shows obvious distribution characteristics. The liquid velocity in the downflow zone is conducive to the liquid that falls to the bottom of the reactor, which increases the circulation times of the liquid in the rising and falling regions. However, the increase in liquid velocity and gas holdup caused by fast gas velocity at the inlet is not conducive to the residence time of reactants in the reactor.

### 3.3. Radiation

The light source is the main energy source of light reaction. Thus, the location of light source and the intensity of light radiation will determine the degradation efficiency of pollutants. Figure 5 demonstrates the radiation intensity of the UV lamp horizontal distributions of the photoreactor. Radiation decreases with the increase in horizontal distance from the lamp as a result of water absorption (UVT = 88%). The position of the draft tube in the reactor not only hinders the ultraviolet radiation from the light source but can also cause an internal circulation effect in the reactor. Therefore, in this study, the location of draft tube in the reactor and the distance from the light source are appropriate.

### 3.4. Chemical Reaction

The prediction of airlift photoreactor performance was simulated on the basis of the CFD model resolution, which includes the specific kinetics for the chemical reaction in the mass balance of the species. Using the dependence of the hydrodynamic distribution of the airlift photoreactor, the kinetic responses of the UV-$H_2O_2$-MB in Table 1 are represented in Figures 6–8 as the chemical reaction results calculated by UDF. Figures 6–8 show the contours and radial distribution of $H_2O_2$, ·OH and MB near the reactor outlet (Section 2 in Figure 1) at different gas velocities, respectively. The simulation of this study is the concentration profile after running from an unsteady state environment to a steady state one.

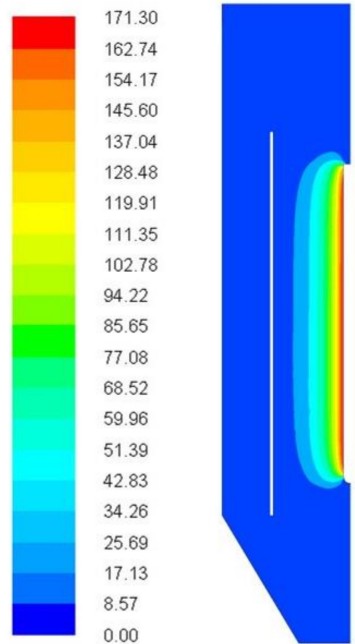

**Figure 5.** Local values of ultraviolet radiation dose calculated in the whole reaction zone of photoreactor $(w/m^2)$.

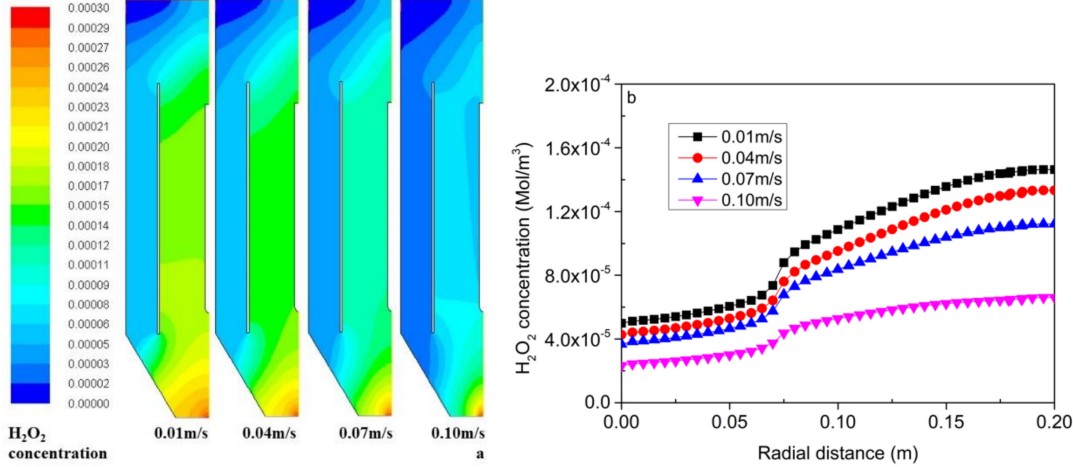

**Figure 6.** The contours (**a**) and the radial distribution (**b**) of molar concentration of $H_2O_2$ at different gas velocity in Section 2.

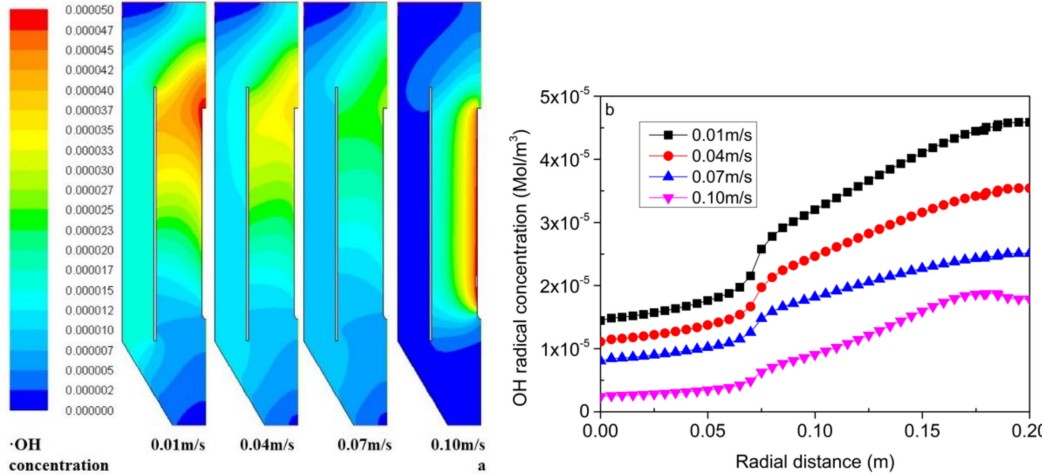

**Figure 7.** The contours (**a**) and the radial distribution (**b**) of molar concentration of ·OH at different gas velocities in Section 2.

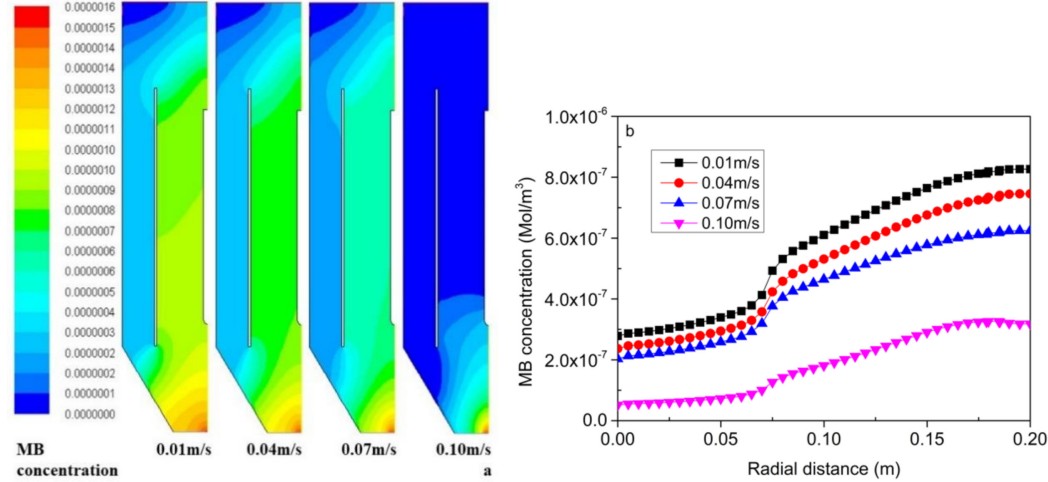

**Figure 8.** The contours (**a**) and the radial distribution (**b**) of molar concentration of MB at different gas velocities in Section 2.

The distribution of hydrogen peroxide concentration at different gas velocities can be seen in Figure 6a. The hydrogen peroxide in this study will enter the bottom of the reactor continuously and quantitatively. With the increase in gas velocity, the hydrogen peroxide concentration in the liquid rising zone near the UV lamp in the reactor showed a decreasing trend, which could reach the level of $5.0 \times 10^{-5}$ mol/m$^3$ (Figure 6b), whereas the maximum concentration was maintained at $1.5 \times 10^{-4}$ mol/m$^3$ under the condition of minimum gas velocity (Figure 6b). The reason is that the hydrogen peroxide near the light source can absorb more ultraviolet rays and generate hydroxyl radicals under the driving of higher gas velocity, thereby leading to the rapid decrease of its own concentration. The concentration contours of hydrogen peroxide depict that the higher the gas velocity in the rising zone is, the more favorable it is for the liquid in the reactor to cross the draft tube and enter the downflow zone to increase the number of cycles. Additionally, they are conducive to the more ultraviolet radiation of various chemical components in the liquid to promote the reaction process. However, the hydrogen peroxide on the other side away from the ultraviolet ray, that is, in the downflow region, also maintains a certain concentration (Figure 6b), indicating that hydrogen peroxide does not completely absorb ultraviolet light to generate free radicals under the specified gas velocity, thereby causing a part of waste to the reaction system.

The distribution of hydroxyl radical concentration at different gas velocities can be seen from Figure 7a. In this reaction system, the formation of hydroxyl radicals is mainly generated by hydrogen peroxide through UV irradiation. With the increase in gas velocity, the concentration of hydroxyl radical in the liquid rising zone near the UV lamp in the reactor showed a decreasing trend, which could reach $1.5 \times 10^{-5}$ mol/m$^3$ (Figure 7b), whereas the highest concentration was maintained at $4.5 \times 10^{-5}$ mol/m$^3$ under the condition of minimum gas velocity (Figure 7b). According to the radial distribution of the contour pictures (b), the hydroxyl radical near the light source is higher, and the concentration is lower with the distance from the light source. The reason is that the hydrogen peroxide near the light source absorbs more ultraviolet light and decomposes more hydroxyl radicals. According to the vertical distribution of contour pictures (b), the hydrogen peroxide that enters the reactor from the inlet gradually absorbs ultraviolet light to produce hydroxyl radicals, and the active hydroxyl radicals will react with the pollutants (MB) in the liquid in a short time period and show a decreasing trend until the outlet is reached. Moreover, the peak concentration tends to move backward because of the influence of chemical reaction and rising zone. The hydroxyl radicals in the advanced oxidation reaction cannot only be generated by the decomposition of hydrogen peroxide but can also be consumed by the degradation of organic pollutants in the reaction system to maintain a certain concentration level in the two directions of generation and consumption.

The distribution of methylene blue concentration at different gas velocities can be seen from Figure 8a. The vertical distribution of contour pictures shows that MB first enters the reactor quantitatively from the inlet of the bottom of the reactor. Thus, MB has the highest concentration. However, the concentration of MB decreases gradually when it is degraded by the hydroxyl radical generated by hydrogen peroxide under ultraviolet radiation in the rising region, indicating that the concentration distribution of pollutants in the reactor can be simulated by CFD numerical method effectively. Almost no MB was observed at the outlet of the reactor, indicating that the degradation reaction was completely degraded within the effective residence time. With the increase in gas velocity, the MB in the rising zone of the reactor showed a decreasing trend that could reach the level of $3.2 \times 10^{-7}$ mol/m$^3$ (Figure 8b), whereas the maximum concentration was kept at $8.3 \times 10^{-7}$ mol/m$^3$ under the condition of minimum gas velocity. The concentration of MB in the reaction system is mainly reduced by the degradation of hydroxyl radicals, and the best degradation effect is obtained when the gas velocity is 0.10 m/s, indicating that in this reactor, the speed and times of liquid circulation in the reactor accelerated with the increase in gas velocity, thereby providing the necessary requirements for the full reaction of organic pollutants and hydroxyl radicals in liquid. The unreacted hydroxyl radicals in the rising zone can be lowered again to the inlet and re-enter the internal circulation system for further water purification. In summary, the contact opportunity for the material components in the liquid varies under different flow conditions. The hydrogen peroxide, hydroxyl radical, and MB that enter the reactor showed a decreasing trend with the increase in contact opportunities. This finding shows that the effective contact opportunities will provide sufficient reaction time for various chemical components in the liquid. However, providing a longer residence time by adjusting the flow rate is not scientific enough. Therefore, the gas flow rate of 0.10 m/s is the best condition for water purification.

### 3.5. Comparison of Experimental and Simulation Results

The methylene blue solution was treated in an internal airlift circulation photoreactor. The comparison of degradation effect and numerical simulation results using continuous experiment under four different gas velocity conditions is shown in Figure 9. The overall removal rate of MB is also generally improved with the increase in gas velocity. At the gas velocity of 0.10 m/s, the numerical simulation results can reach 93.34%, and the highest removal rate of the actual experiment is 69.87%. When the minimum gas velocity is 0.01 m/s, the removal rate of numerical simulation is 87.61%, and the highest removal rate is 43.70%. This finding shows that the reactor structure is reasonable and can play the advantages of photocatalytic reaction effectively to improve the degradation efficiency of MB. The experimental data are generally lower than the numerical prediction data, and the difference

range is approximately 34%. With the increase of gas velocity, the standard deviation of repeated test data is 2.47, 4.28, 4.52, and 7.91, respectively. This is because the larger turbulence intensity can increase the mixing effect of material components, but it has a certain impact on the effluent quality. Generally, we still obtained a good prediction result. Some kinetic parameters involved in the reaction are not considered in the simulation because this numerical simulation is mainly aimed at the distribution of $H_2O_2$, $\cdot$ OH, and MB in the liquid phase.

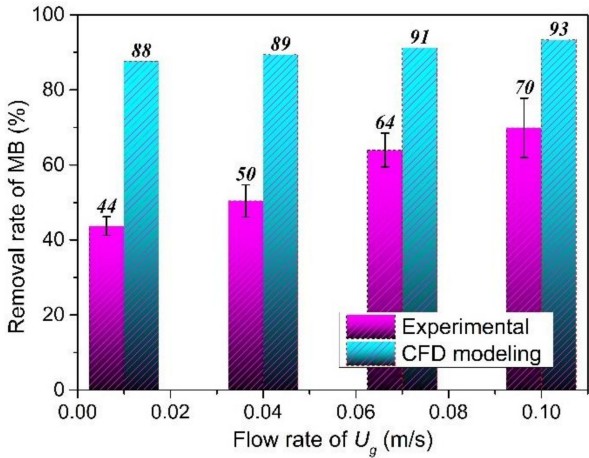

**Figure 9.** Comparison of experimental data and CFD results.

## 4. Conclusions

In this paper, the Euler model was used to simulate the liquid flow field distribution, gas volume fraction, and turbulence intensity in an internal airlift circulation photoreactor. Then, the distribution of the main chemical components in the reactor under the condition of photocatalytic reaction was simulated by using the reaction kinetics model. Finally, the results were compared with the experimental data. The simulation results of the flow field show that the liquid under the driving of gas velocity can result in the remarkable circulation flow in the reactor, improve the mixing effect effectively, increase the contact area among substances, and provide favorable conditions for improving the photocatalytic degradation effect. The model used in this study can also describe the chemical reaction behavior in the internal airlift circulation photoreactor accurately, and the predicted results are similar to the actual experimental data, thereby providing reference for the establishment of the chemical reaction model of this kind of reactor. This model can also provide theoretical support for the design and scale-up of large-scale airlift inner loop photoreactor.

**Author Contributions:** Conceptualization, T.J. Data curation, M.L., F.Z. and T.J. Formal analysis, M.L. Funding acquisition, M.L. Investigation, T.J. and Q.G. Methodology, M.L., F.Z., T.J. and G.W. Project administration, M.L Resources, M.L. Software, M.L. Supervision, F.Z. Validation, F.Z. Visualization, Q.G. Experiments, M.L. Writing-original draft, T.J. Writing-review & editing, G.W. All authors have read and agreed to the published version of the manuscript.

**Funding:** This work was funded by Natural Science Research of Jiangsu Higher Education Institutions of China (No.19KJB610012, No.18KJB610006), and Introduction Talent Scientific Research Foundation Project of NanJing Institute of Technology (No. YKJ201847) and Supported by the Cooperation Fund of Energy Research Institute, Nanjing Institute of Technology (No. CXY201925).

**Conflicts of Interest:** The authors declare no conflict of interest.

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
