# Peer review of "CFD Modeling of UV/H2O2 Process in Internal Airlift Circulating Photoreactor"

_water, doi:10.3390/w12113237_

Round 1

Reviewer 1 Report

The manuscript investigated an AOP process of methylene blue removal in an internal airlift circulating photoreactor, using both CFD modeling and an experimental approach. The introduction needs to be revised to detail the uniqueness of UV/H2Ophotolysis in the internal airlift circulating photoreactor and research gaps in CFD modeling of such process. The kinetic model of UV/H2O2 advanced oxidation process of methylene blue removal seems using old data (before 1990) in Table 1 for numerical model development, while newer development should be adopted (e.g., Alpert, S. M., Knappe, D. R., & Ducoste, J. J. (2010). Modeling the UV/hydrogen peroxide advanced oxidation process using computational fluid dynamics. water research, 44(6), 1797-1808.), or authors should validate the selection of such kinetic equations. The CFD results were merely described, and further discussions are encouraged. Overall, the manuscript needs a major revision before publishing. 

Specific comments: 

Line 27, not sure what is "re pollution"? 

Line 45, check the unit of reaction rate constant.

Line 66, citation missing. 

Line 149, why choosing r/R 0.4? Is it worthing to investigate the impact r/R on system performance? 

Line 231-232, If increased turbulence intensity reduces the residence time, why not further investigate the impact of residence time under different air velocities? 

Line 320, the maximum concentration is below 1.0×10-6 in Figure 8. How the authors obtain a value of 8.5×10-5?

Line 347, what's the standard deviation of the experimental data? 

Author Response

We are thankful to the reviewer 1 for making constructive and useful comments/suggestions which enable us to improve the quality of our manuscript. Every comments/suggestions were responded herewith.

The manuscript investigated an AOP process of methylene blue removal in an internal airlift circulating photoreactor, using both CFD modeling and an experimental approach. The introduction needs to be revised to detail the uniqueness of UV/H2Ophotolysis in the internal airlift circulating photoreactor and research gaps in CFD modeling of such process.

Response: The uniqueness of UV/H2Ophotolysis in the internal airlift circulating photoreactor and research gaps in CFD modeling of such process were added as shown in Line 63-71.

The kinetic model of UV/H2O2 advanced oxidation process of methylene blue removal seems using old data (before 1990) in Table 1 for numerical model development, while newer development should be adopted (e.g., Alpert, S. M., Knappe, D. R., & Ducoste, J. J. (2010). Modeling the UV/hydrogen peroxide advanced oxidation process using computational fluid dynamics. water research, 44(6), 1797-1808.), or authors should validate the selection of such kinetic equations. The CFD results were merely described, and further discussions are encouraged. Overall, the manuscript needs a major revision before publishing. 

Response: We are thankful for the reviewer’s comment and suggestion. We have read a lot of authoritative papers on this field, all of which are based on the data of kinetic parameters made by predecessors for a long time, and there is no new progress in this field in recent years. However, according to the reviewer's advice, we also changed all the reference papers to 2010 papers (Alpert, S. M., Knappe, D. R., & Ducoste, J. J. (2010). Modeling the UV/hydrogen peroxide advanced oxidation process using computational fluid dynamics. water research, 44(6), 1797-1808.). The CFD results were rewritten, and further discussions were added as shown in “3.5. Comparison of Experimental and Simulation Results”.

Specific comments: 

Q1. Line 27, not sure what is "re pollution"? 

Response 1: The use of “re pollution” here is not appropriate, the whole paragraph was re-edited. The changed statement is “Advanced Oxidation Processes (AOPs) is a cost-effective solution for removal of harmful and refractory pollutants from water”(Line 27-29).

Q2. Line 45, check the unit of reaction rate constant.

Response 2: The unit of reaction rate constant was changed to M-1s-1. (Line 48).

Q3. Line 66, citation missing. 

Response 3:The citation was added. (Line 92).

Q4. Line 149, why choosing r/R 0.4? Is it worthing to investigate the impact r/R on system performance? 

Response 4 The r/R value was choosed  based on the results of our previous studies (Minghan Luo, Qiuwen Chen, taeseop Jeong et. 2017 numerical modeling of a three-phase internal air lift circulating photoreactor. Water SCI technology. 76 (11), 3044-3053). The position of the draft tube in the airlift reactor is very important, because after the gas enters the reactor from the bottom, the size of the inner diameter of the draft tube (i.e. the ratio of the inner diameter of the draft tube to the inner diameter of the reactor) affected the liquid density in the draft tube, and then the velocity and liquid distribution between the up flow and the down flow were affected. So it is very necessary.

Q5. Line 231-232, If increased turbulence intensity reduces the residence time, why not further investigate the impact of residence time under different air velocities? 

Response 5:Turbulence intensity shows mixing characteristics, which is independent of residence time. The description used before was not correct, so this paragraph has been deleted in this article. (Line 266-267)

Q6. Line 320, the maximum concentration is below 1.0×10-6 in Figure 8. How the authors obtain a value of 8.5×10-5?

Response 6As the reviewer said, the maximum concentration in Figure8(b) should be 8.3×10-7 rather than 8.5×10-5,this is due to the author’s negligence and has been modified.  (Line354-355).

Q7. Line 347, what's the standard deviation of the experimental data? 

Response 7With the increase of gas velocity, the standard deviation of repeated test data is 2.47, 4.28, 4.52 and 7.91, respectively. They were added in Line 379-380.

Reviewer 2 Report

This problem is relevant for journal scope. The manuscript is well written, I could not find typing errors. The manuscript follows the formal regulations of journal. The figures are informative and well-structured.

I suggest the acceptance after minor revision.

Remarks, suggestions

  1. Please cite more papers from this journal at the last two years in the similar topic of this research.
  2. Please add Nomenclature part to the manuscript.
  3. Please describe the applied spectrophotometry method in detail.
  4. Please introduce the experiment apparatus.
  5. Please add the error bars to the graphs where is it possible.
  6. Please explain the cause of difference in Figure 9 in detail.

Author Response

We are thankful to the reviewers for making constructive and useful comments/suggestions which enable us to improve the quality of our manuscript. Every comments/suggestions were responded herewith.

Remarks, suggestions:

Q1. Please cite more papers from this journal at the last two years in the similar topic of this research.

Response 1Papers from this journal at the last two years in the similar topic of this research were added. (Line 418 & 423).

Q2. Please add Nomenclature part to the manuscript.

Response 2 In order to further analyze the data of various physical elements in the reactor, the nomenclature part in this study mainly includes section 1 and section 2, as well as the r / R ratio. The naming of this part has been added to the manuscript. (Lien 183 - 185).

Q3. Please describe the applied spectrophotometry method in detail.

Response 3The absorbance of solution at 664 nm was monitored to determine the concentration of MB by a UV spectrophotometric probe (UV1800, Shimadzu Co., Japan spectrophotometer). The MB standard solution with concentration of 0.1mg/L、0.2mg/L、0.3mg/L、0.4mg/L、0.5mg/L were prepared. The absorbance of the solution is measured at 664nm by UV visible spectrophotometer. The standard curve of MB solution is obtained by taking the absorbance as abscissa and MB concentration as ordinate. The relationship between absorbance and concentration of MB solution can be described by y = 0.189x + 0.003, where R2 = 0.9998. Finally, the absorbance of the samples taken at different times is measured, then the concentration value is calculated by the above equation.(Line 226-227).

Q4. Please introduce the experiment apparatus.

Response 4:The experiment apparatus was shown in Fig. 1, the inner part of the reactor is composed of a 60cm long draft tube and a 50cm long UV lamp, and the ratio of the draft tube and the reactor (r/R) is 0.4. When the reactor is running, the solution with MB and H2O2 was injected from the bottom of the reactor at a rate of 0.3m/s continuously. At the same time, the air was injected at flow velocity of 0.01, 0.04, 0.07, and 0.10 m/s, respectively. The reactor reached a stable condition after 120 min, then a total of 6 samples were taken at 30 min time intervals for methylene blue determination. (Line 178 - 182).

Q5. Please add the error bars to the graphs where is it possible.

Response 5The error bars were added to the Fig. 9.

 Q6. Please explain the cause of difference in Figure 9 in detail.

Response 6:Some free radicals were generated during the experimental process, which have an adverse effect on the degradation of MB. While the effect of these radicals were not considered in the numerical simulation.

Round 2

Reviewer 1 Report

The manuscript is in good shape for publication. 

Reviewer 2 Report

Thank you very much the answers.

In my opinion the answers are professionally well-founded. I suggest the acceptance in this present form for publication.